# Behavioral, electrocorticographic and electrocardiologic changes in *Colossoma macropomum* (Tambaqui) in the effect of cunaniol

**Akira Otake Hamoy**[1], **Suzane Maia da Fonseca**[1], **Giovanna Lourenço Cei**[1], **Fábio Leite do Amaral Júnior**[1], **Maria Klara Otake Hamoy**[1], **Rafaela Marques Ribeiro**[1], **Luis Andre Luz Barbas**[2], **Nilton Muto**[3], **Moisés Hamoy**[1] *

1 Laboratory of Pharmacology and Toxicology of Natural Products, Biological Sciences Institute, Federal University of Para, Belem, Para, Brazil, 2 Tropical Species Aquaculture Laboratory, Federal Institute of Education, Science and Technology of Pará (IFPA), Castanhal Campus, Castanhal, Brazil, 3 Center for the Valorization of Bioactive Compounds, Institute of Biological Sciences, Federal University of Pará, Belém, PA, Brazil

* hamoyufpa@gmail.com

## Abstract

The *Clibadium* spp. is a shrub of occurrence in the Amazon, popularly known as Cunambi. The compounds in the leaves demonstrate ichthyotoxic properties, and its major substance, cunaniol, is a powerful central nervous system stimulant with proconvulsant activity. Few current studies relate behavioral changes to the electrophysiological profile of fish poisoning. This study aimed to describe the behavioral, electromyographic, electroencephalographic, electrocardiographic, and seizure control characteristics of anticonvulsant drugs in *Colossoma macropomum* submitted to cunaniol intoxication during bathing containing 0.3 μg/L cunaniol. The behavioral test showed rapid evolution presenting excitability and spasms, which were confirmed by the analysis of Electroencephalogram (EEG), Electromyogram (EMG), and changes in cardiac function detected in the ECG. Cunaniol-induced excitability control was evaluated using three anticonvulsant agents: Phenytoin, Phenobarbital, and Diazepam. While phenytoin was not effective in seizure control, diazepam proved to be the most efficient. These results demonstrate the susceptibility of *Colossoma macropomum* to cunaniol poisoning, given that the central nervous system and electrocardiographic changes were considered severe.

## Introduction

Plants of the genus *Clibadium* spp. are popularly known as cunambi are abundantly found in Latin America, mainly in northern and northeastern Brazil [1]. Leaves of this shrub are widely used for predatory fishing, due to their ichthyotoxic properties. The main effects of cunaniol on fish behavior were reported by Quilliam & Stables (1968) [2]. In addition to this ichthyotoxic effect, macerated leaves are used in popular medicine to treat erysipelas, and hemorrhages, as anti-inflammatories, and as a natural insecticide [3].

**Data Availability Statement:** All relevant data are within the paper and its Supporting Information files.

**Funding:** The author(s) received no specific funding for this work.

**Competing interests:** The authors have declared that no competing interests exist.

In relation to its chemical composition, cunaniol ($C_{14}H_{14}O_{2)}$) has as nomenclature: (2R, 3S) -2 - [(Z) -non-1-en-3,5,7-triinyl] oxan-3-ol] following IUPAC (International Union of Pure and Applied Chemistry) rules. It presents a polyacetylene alcohol group that can act as a negative modulator of the GABAA receptor function or as a positive modulator of the GABAA receptor function, particularly those that contain b2 subunits [4].

The seizure effects of cunaniol in fish continue to be researched for their behavioral changes in animals. Costa et al (2006) [5] analyzed the convulsive properties of *Clibadium surinamensis* extract in Swiss mice and evaluated the excitability of the central nervous system (CNS) caused by the substance. According to Hamoy and colleagues (2018) [6], the treatment with cunaniol in Wistar rats resulted in modifications of the EEG, proving the potent action of this extract as a chemoconvulsant agent with changes in Beta brain oscillations (12 to 28 Hz).

Costa (2006) reported that the molecular mechanism of the CNS excitation caused by oral administration of *Clibadium surinamensis* extract is not unknown but is blocked by increased activity of GABAA. Still, according to Costa (2006), sodium channels are not directly involved in the toxic activity of the plant, and other possible mechanisms of convulsive effect are also considered, such as the blocking of $K^+$ channels.

The study of antiepileptics to evaluate the blockade of cunaniol-induced seizures should also be highlighted, although little has been elucidated. Of the studies analyzed, diazepam has a better control as a seizure blocker when compared to phenytoin and phenobarbital [5, 6]. As an underexplored subject, a range of investigative possibilities coexist regarding the cunaniol action mechanism, and the use of these drugs might raise hypotheses for its understanding. Thus, this study aims to analyze behavioral and electrophysiological responses in Tambaqui (*Colossoma macropomum*) induced by cunaniol.

## Materials and methods

### Experimental animals

The fish used were young forms of tambaqui, *Colossoma macropomum*, purchased from a fish breeding farm. The animals (20.9 ± 1.5g) were stored in aquariums in the Experimental Biotery of the Pharmacology and Toxicology Laboratory of Natural Products of the Federal University of Pará (UFPA) in a controlled temperature environment (25 to 27.6˚C) and photoperiod 12 h L: 12 h D. Feeding was provided twice daily with commercial ration (32% protein) until satiety. Concomitantly with the daily siphoning for the removal of uneaten food and feces, the water was partially renewed (approximately 20% of the volume of the tanks) with water from the same source. During acclimation (15 days), water quality variables such as water temperature (˚C); hydrogen potential (pH); dissolved oxygen (DO); ammonia ($NH_3^+$), and total hardness were monitored and maintained as follows: Temp. 26.8˚C; pH 7.5; DO> 5.0 mg/L; $NH_3^+$ 0.1 mg/L and total hardness 70 mg/L $CaCO_3$. A veterinarian (Moisés Hamoy, License CRMV PA1099) monitored (twice a day) the overall clinical health status of the fish batch before and throughout the experiments.

After the trials, all animals (a total of 120 fishes) were euthanized upon administration of propofol at a high dose (30 mg/kg i.p.), as advised by the Brazilian National Council for the Control of Animal Experimentation (CONCEA/Brazil). The decision to euthanize right after the electrophysiological measurements was made by the veterinarian and based on the poor prognosis after the experiment-related handling, surgery and the inexistence of appropriate protocols should they be necessary for analgesia or treatment of potential infections in tambaqui fish. Any suffering that could arise from late convulsions post-exposure to cunaniol, severe inflammation and infection following the removal of the electrodes, or any other pharmacological distress after the experiments could therefore be avoided.

The experiment and all procedures used herein were previously approved by the UFPA Animal Experimentation Ethics Committee: CEUA-UFPA Bio 0101–12.

## Cunaniol extraction

Botanical samples of *Clibadium* spp. were collected in the countryside of Castanhal—PA—Brazil, in a dry land area. Cunaniol was obtained from leaves of the *Clibadium surinamense* at the Chromatography Laboratory (Institute of Chemistry–Federal University of Para) following the International Guidelines suggested by the World Health Organization (WHO, 2003). The Clibadium tree (known by its popular name Cunambi) has dried samples (exsiccates) in the West Amazon EMBRAPA'S IAN Herbarium (Belém-Pará-Brazil) under registration # 183939.

Cunaniol was handled and protected from light exposure, as it was observed to be a light-sensitive molecule. It was diluted in 0.1% Tween 20 (Sigma-Aldrich) to a concentration of 5 mg/mL and stored at– 20°C, and then diluted to 0.3 μg/L in aquarium water.

## Animal groups

Tambaqui juveniles (20.9 ± 1.5g) (120 subjects) were randomly assigned to each EEG and EMG group (n = 12), behavioral and ECG group (n = 9). All groups had contact for 10 minutes with cunaniol (0.3 μg/L) diluted in aquarium water (2000 mL), identified in the following treatments: a) behavioral control group (n = 9) with individuals who had no contact with cunaniol; b) behavioral treated-group (n = 9), with behavioral evaluation during 10 minutes of contact with cunaniol; c) the EEG control group (n = 12) for the electroencephalographic recording group (EEG) composed of animals without contact with cunaniol; d) electroencephalographic recording group (n = 12) during contact with cunaniol; e) EMG control group (n = 12) for electromyographic recording (EMG); f) Electromyographic recording group (n = 12) during contact with cunaniol; g) ECG control group (n = 9) for electrocardiogram (ECG); h) Electrocardiographic recording group (n = 9) in the presence of cunaniol; (i) Diazepam anticonvulsant test group (n = 9); j) Phenytoin anticonvulsant test group (n = 9); l) Phenobarbital anticonvulsant test group (n = 9).

For "i, j, and l" groups, 10 mg/kg of diazepam, phenytoin, and phenobarbital were administered intraperitoneally (IP) ten minutes before contact with cunaniol (0.3 μg/L) and, meanwhile, a 10-minute record was taken to assess the reduction in the onset of central excitability.

## Behavioral pattern during contact with cunaniol

Through empirical observation and literature review on behavior analysis, it was adapted to the work, providing classification and description of behaviors, to systematize the results obtained in the experiment upon exposure to 0.3 μg/L cunaniol.

In this study, the following behaviors of *Colossoma macropomum* exposed to cunaniol were verified taking into consideration the order of appearance of the behavior:

**Freezing.**   The animal stands still. (Freezing)

**Excitability (Seizure).**   fast movements, swimming without direction, without loss of posture reflex (Circular swimming, horizontal impulses, and rotation around the axis may be present).

**Loss of Posture Reflex.**   The animal cannot maintain its overall pattern of motor activity (back-ventral), sometimes assuming a lateral or ventral-dorsal posture.

**Muscle Spasms.**   decreased normal swimming activity with muscle spasms that become frequent and can propel the fish forward.

## Confection and electrode implantation in fish

In this study, the method of Barbas et al. (2021) [7] was used. The electrodes (record and reference) were made with two identical stainless-steel segments measuring 10 mm in length and 0.5 mm in radius (GN INJECTA IND. E COM. LTDA, São Paulo Brazil). Isolated and conjugated with a thin layer of epoxy. The electrodes are conjugated at a distance of 2 mm.

The animals were kept out of the water and fixed in sponges, under a continuous flow of 30 ml of a benzocaine solution at 80 mg/L for anesthesia maintenance. For electroencephalograms, the recording was made in the midbrain region, with electrodes being introduced into the brain along the sagittal line (longitudinal) and caudal edge of the eye (transverse). The intersection location between the lines corresponds to the reference point as ground zero. From this point, they were implanted with 1 mm to the right and 1 mm to the left, after the previous drilling with the dental appliance, conditioned and subsequently fixed with self-curing acrylic. The right side is used as a record and the left side as a reference. After electrode placement, the animals were kept for 48 hours in a recovery aquarium for complete anesthetic metabolism and after preparation, they were placed in an aquarium inside a Faraday cage for recording purposes.

For electromyographic recording, conjugated electrodes were introduced 5.0 mm into the muscle below the dorsal fin, likewise, the animal was placed in aquariums inside Faraday's cage and subjected to ten minutes of contact with a solution containing 0.3 μg/L of cunaniol.

For cardiac monitoring, the electrodes were constructed of non-welded stainless steel rods with a diameter of 0.3 mm and 5.0 mm in length. The position of fixation of the reference electrode followed the indication of the cardiac vector, being fixed in the ventral part, 0.2 mm after the end of the opercular cavity, and the recording electrode was inserted 2.0 mm below the pectoral fin. Heart rate (BPM), the amplitude of record (mV), QRS duration (S), R-R interval (S), and Q-T duration (S) (Barbas, 2017) were analyzed from the records.

## Equipment used to record and data acquisition

The record select rods were connected to one high-impedance amplifier (Grass Technologies, P511) with amplification in 2000x. The voltage records were low-pass filtered at 0.3 kHz and high-pass filtered at 0.3 Hz and monitored by an oscilloscope (Protek, 6510). The data were continuously digitalized and stored at 1kHz sampling resolution at a computer equipped with a board of data acquisition (National Instruments, Austin, TX) stored in a computer hard drive for offline analysis by software (LabVIEW express).

## Data analysis

For analysis of the acquired signals, a tool was developed using the Python programming language version 2.7. The NumPy and SciPy libraries were used for the mathematical processing and the matplotlib library for the graphs. The graphic interface was developed using the PyQt4 library.

The amplitude graphs are intended to show the potential difference between the electrodes of reference (left hemisphere) and the record (right hemisphere). In the signal, 1000 samples per second were observed. The spectrograms were calculated using a Hamming window with 256 points (256 / 1000 seconds), and each frame was generated with an overlap of 128 points per window. For each frame, the power spectral density (PSD) was calculated by Welch's average periodogram method. The frequency histogram was generated by first calculating the PSD of the signal using the Hamming window with 256 points without overlap, with the resulting PSD a histogram was assembled with bins of 1 Hz. To analyze the difference between the experiments, a graph with the mean and standard deviation of PSD of several experiments was

assembled, each wave of the graph was generated from a set of experiments, where the individual PSD was calculated and the mean and standard deviation of each group is shown, for calculation of PSD Hamming window of 256 points without superposition was used.

## Statistical analysis

After verifying compliance with the assumptions of normality and homogeneity of variance, through the Kolmogorov-Smirnov and Levene tests, respectively, comparisons of the EEG mean power values were made by one-way ANOVA, followed by the Tukey test. For the analysis of the groups for the electrocardiograms, the Student $t$-test, and the non-parametric test were used. GraphPad Prism® 8 software was used for the analysis. The minimum significance level was set at $*p < 0.05$, $**p < 0.01$, and $***p < 0.001$ was considered statistically significant in all cases.

## Results

No mortalities or disease occurred during acclimation or throughout the experimental period.

### Behavioral patterns observed in tambaqui juveniles after contact with cunaniol

During the behavioral characterization experiment, four behaviors were observed that were repeated in all tests and obeyed a chronological order of appearance, initially, the immobility behavior presented a latency of 6.444 ± 4.187 seconds which corresponds to the organism's first reaction to the exposure site. Behavior evolution occurs rapidly with the animal presenting excitability triggered by a convulsive condition, with an onset latency of 26.00 ± 22.66 seconds. Then there is a loss of posture reflex (64.22 ± 12.50 seconds) and muscle spasms with forwarding impulses (80.11 ± 14.07 seconds). This behavior after its onset was maintained until the end of the 10 minutes of observation.

### Cunaniol caused an increase in the EEG amplitude of tambaqui

The animals that did not come into contact with the cunaniol showed uniformity in the EEG, which remained at low amplitude and with more intense energy distribution at frequencies below 10 Hz, as can be seen in Fig 1A. After contact with the cunaniol, the tambaqui juveniles presented alterations in the electroencephalographic records, and two phases of records called potentials bursts can be detected, in which there is an increase in frequency and amplitude with an increased energy distribution up to 50 Hz (Fig 1B), and isolated spikes, with a decrease in the intensity of amplitude and number of shots and energy levels of the record (Fig 1B). The amplitude graph showed that the control had a mean of 0.1139 ± 0.03195 $mV^2$/Hz x $10^{-3}$, and the group that came into contact with the cunaniol presented a mean EEG amplitude of 3.845 ± 1.562 $mV^2$/Hz x $10^{-3}$, which presented statistical difference to the control. When the saving phase of potentials is analyzed, the mean amplitude was 12.12 ± 2.085 $mV^2$/Hz x $10^{-3}$, revealing the component that most increase the amplitude in the register during contact with the cunaniol. In the isolated spikes phase, the mean was 1.740 ± 0.5898 $mV^2$/Hz x $10^{-3}$, which showed a significant statistical difference in the recording phases (Fig 2).

### Cunaniol increased electromyographic record amplitude with reduced contact-dependent activity in tambaqui

In the control animals, they presented muscle contractions compatible with normal swimming in the electromyographic register, which remained at low amplitude and with more intense

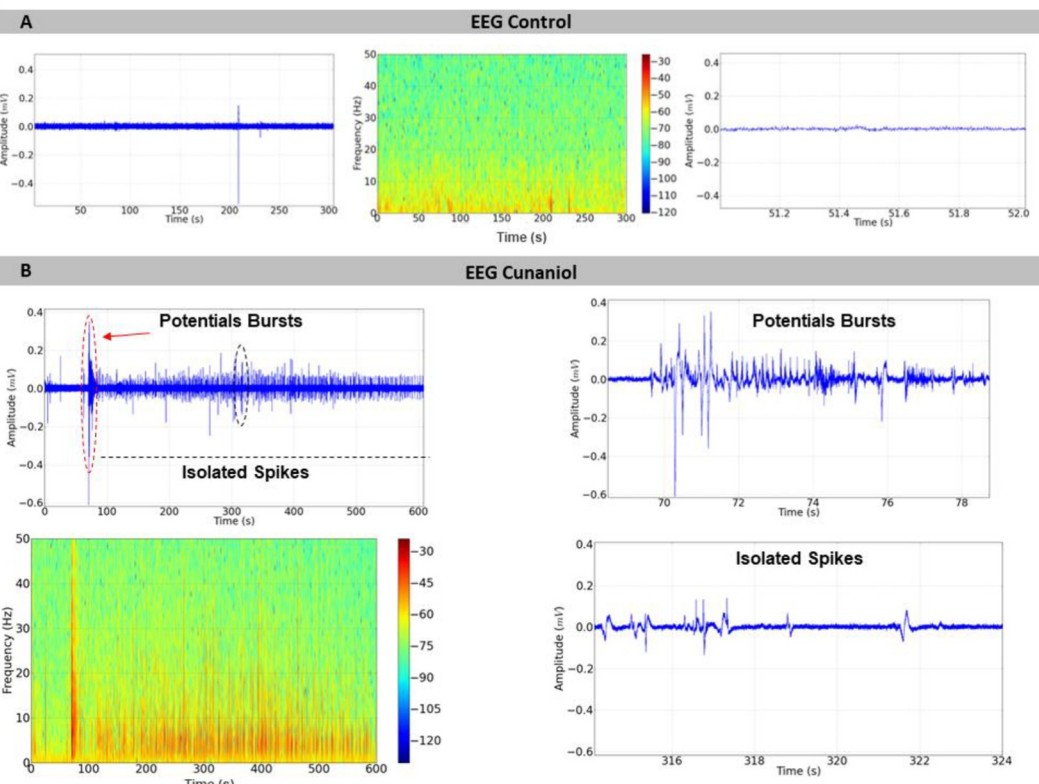

**Fig 1. Demonstration of the electroencephalographic (EEG) tracing of tambaqui (*Colossoma macropomum*) in the control (left), 600-second power distribution spectrogram profile (center), 1s amplification of the control EEG (right).** EEG tracing of the mesencephalic region of tambaqui under the action of cunaniol (0.3 μg/L) lasting 600 seconds (on the left), the record shows changes in amplitude and frequency characterized as potential burst and isolated peaks that were amplified to demonstrate the characteristics (left) and a spectrogram demonstrating the energy distribution during contact (lower left quadrant) (B).

energy distribution at frequencies up to 30 Hz, as can be seen in Fig 3A. After contact with the cunaniol, the tambaqui juveniles presented alterations in the electromyographic records and a period of immobility, excitability (convulsion), and muscle spasms that corroborate the behavioral data observed during excitability, where there is an increase in frequency and amplitude with an increase in energy distribution up to 50 Hz (Fig 3B), and muscle spasms, where there is a decrease in the intensity of amplitude, several shots and energy levels of the record, gradually (Fig 3B). The amplitude graph showed that the control had an average of 16.54 ± 3.096 mV$^2$/Hz x 10$^{-3}$, and the group that came into contact with the cunaniol presented an average EMG amplitude of 33.32 ± 8.785 mV$^2$/Hz x 10$^{-3}$, which showed the statistical difference to the control. When the muscular excitability phase is analyzed from the register, the mean amplitude was 53.78 ± 15.34 mV$^2$/Hz x 10$^{-3}$, the component that most increases the amplitude in the register during contact with the cunaniol. During muscle spasms, the mean was 9.742 ± 2.071 mV$^2$/Hz x 10$^{-3}$. This result showed no statistical difference for the control group (Fig 4).

## Cunaniol alters the cardiac function of tambaqui

Taking into consideration the cardiac normal parameters shown Fig 5A and 5B, we measured cunaniol interference during the final 300 seconds of recording 10 minutes of contact in

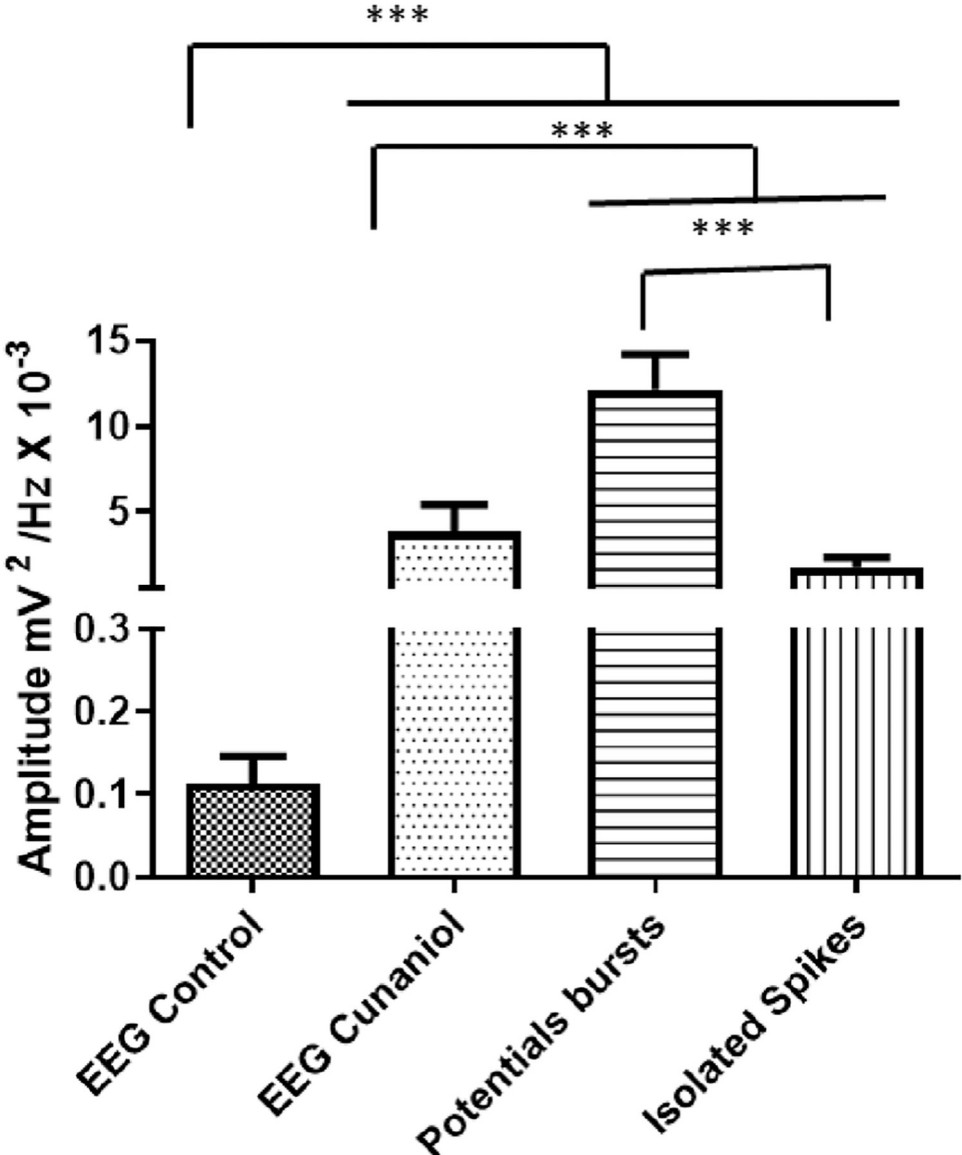

**Fig 2. Mean amplitudes were recorded during the electroencephalogram (EEG) of *Colossoma macropomum* juveniles subjected to contact with 0.3 μg/L cunaniol.** Amplitude analysis in the potential bursts and isolated spikes phases. Recordings were made at 600s duration with frequencies up to 50 Hz. [ANOVA and Tukey's test (n = 12)].

tambaqui due to intense muscle contraction interfering with ECG recording. The *C. macropomum* under cunaniol influence ECG of is shown in Fig 6A and 6B, and it is possible to identify the P wave, QRS complex, and T wave in the tracing.

During the period of contact with cunaniol, there was a significant reduction in heart rate (Fig 6A and 6B), and the control group had a mean frequency of 96.22 ± 5.608 (BPM), which showed a statistically significant difference for the cunaniol group with a mean of 23.11 ± 10.02 (BPM) (Table 1).

For the amplitude of the records, mean of 0.3909 ± 0.07593 mV was observed for the control group, with a statistical difference for the cunaniol group, which had an average of 0.3169 ± 0.03283 mV, showing impairment in the contraction force of the heart. The R-R

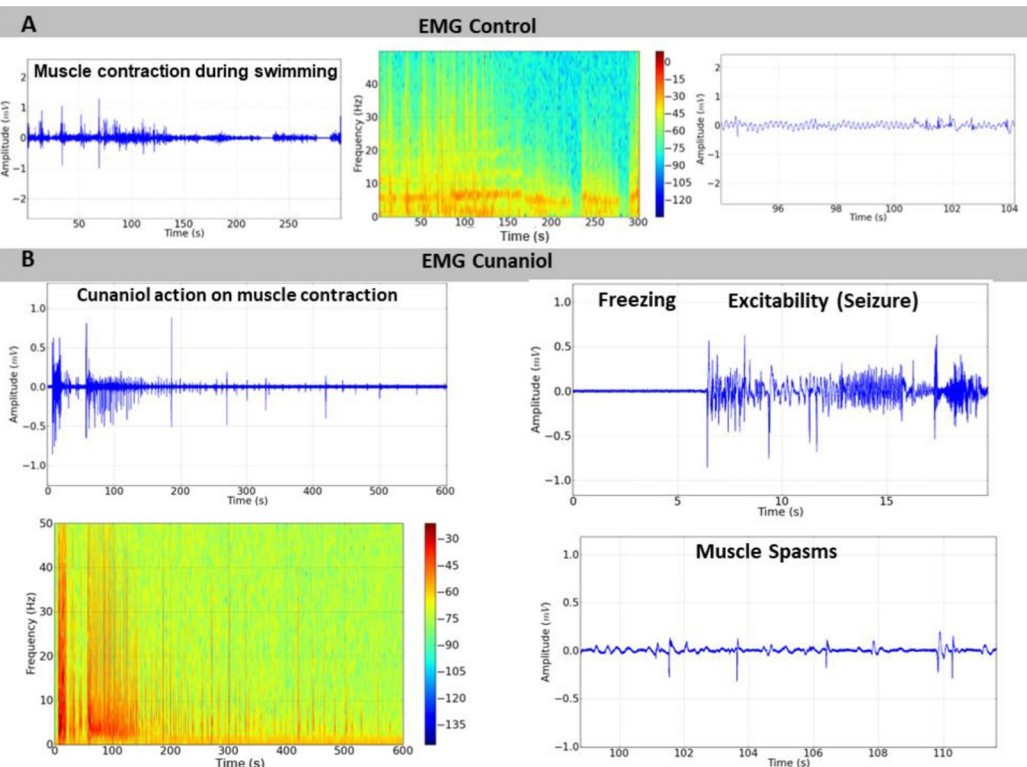

**Fig 3.** Electromyographic (EMG) records of tambaqui, *Colossoma macropomum* at baseline (normal swimming) (A), EMG records performed on animals submitted to 0.3 μg/L bath with cunaniol, characterized as a period of intense contraction and increased energy distribution. In magnification, immobility can be proven followed by excitability and muscle spasms (B). Records made in 600 s.

interval for the control group with a mean of 0.6084 ± 0.02037 seconds presented a statistical difference for the animals that had contact with cunaniol, with a 5.487 ± 1.456 seconds average. The QT interval that represents ventricular depolarization and repolarization presented a mean for the control group of 0.2675 ± 0.04457 seconds and the cunaniol-treated group presented a mean of 0.4271 ± 0.06760 seconds, which shows that contact with cunaniol increases the time for the treatment to occur repolarization of tambaqui heart. Ventricular depolarization time represented by the QRS duration for the control group was 0.03012 ± 0.001767 seconds and for the cunaniol-treated group, the mean was 0.05858 ± 0.008562 seconds, showing a statistical difference for the control group (Table 1). The group that came in contact with the concentration of 0.3 μg/L cunaniol showed a statistical difference for all parameters (*p <0.05, ***p<0.001).

## Diazepam reduces changes in the EEG of the *Colossoma Macropomum* upon contact with cunaniol

The evaluation of anticonvulsant activity to block the onset of cunaniol-induced seizure outbreaks is shown in Fig 7. The first drug to be tested was diazepam, whose electroencephalographic tracing demonstrated little variation and energy distribution in the spectrogram (Fig 7A). When compared to the control record, which presented an average amplitude of 0.1139 ± 0.03195 mV$^2$ / Hz x 10$^{-3}$, the diazepam + cunaniol group with an average amplitude of 0.4472 ± 0.2427 mV$^2$ / Hz x 10$^{-3}$ did not present statistical difference (Fig 8). The phenytoin pretreatment group showed low efficacy to control cunaniol seizures (Fig 7B) and had a mean

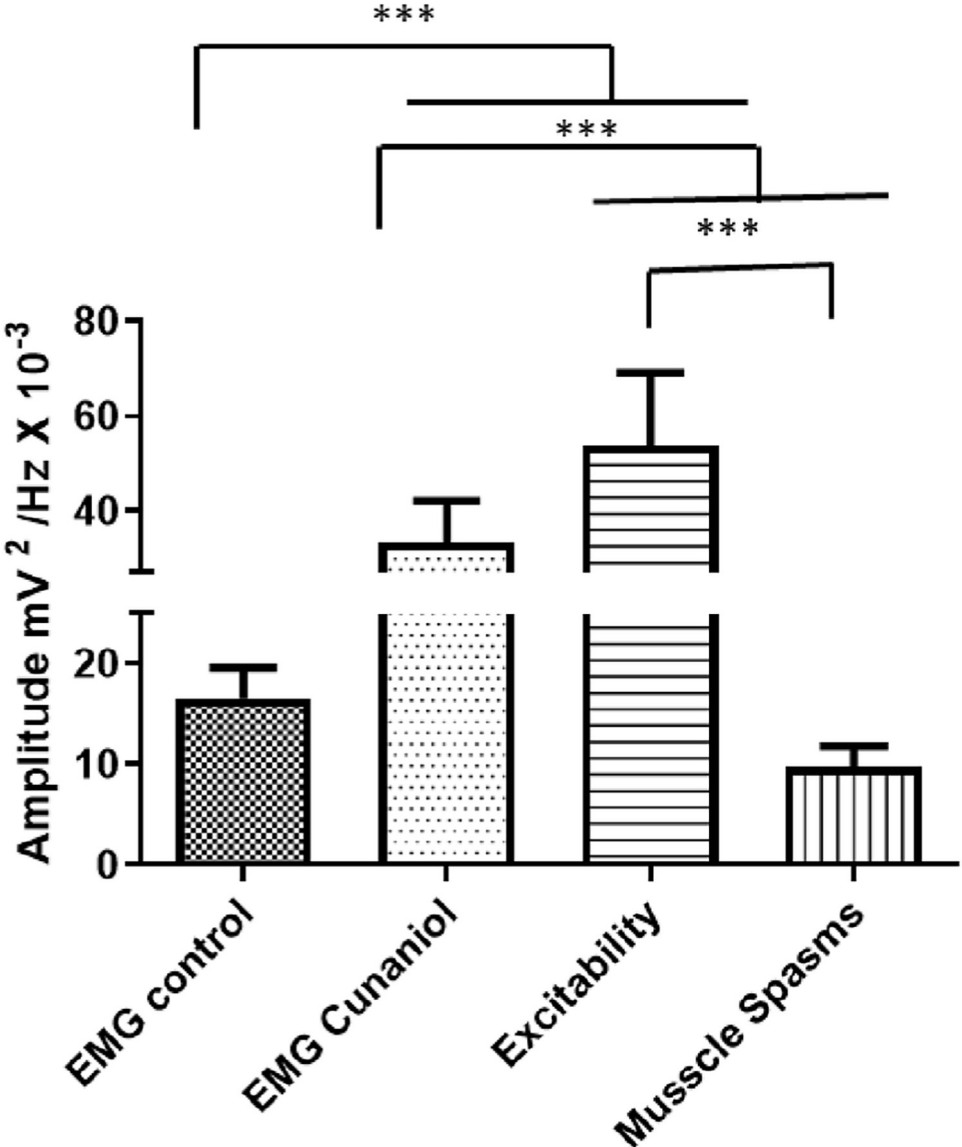

**Fig 4. Mean amplitudes were recorded during the electromyogram (EMG) of *Colossoma macropomum* juveniles subjected to contact with 0.3 µg/L of cunaniol.** Analysis of amplitudes, excitability (convulsion), and muscle spasms. Recordings were made at 600s duration with frequencies up to 50 Hz. [ANOVA and Tukey test (n = 12)].

amplitude of $3.072 \pm 1.131$ mV$^2$ / Hz x $10^{-3}$, showing no statistical difference for the cunaniol group with an amplitude of $3.845 \pm 1.562$ mV$^2$ / Hz x $10^{-3}$ (Fig 8). After phenobarbital application to prevent cunaniol-induced seizure outbreaks, changes in the EEG recordings observed in Fig 7C were observed. The mean amplitude observed in the recordings was $1.656 \pm 0.4438$ mV$^2$ / Hz x $10^{-3}$ and demonstrated statistical differences between the control and cunaniol groups (Fig 8).

## Discussion

Behavioral alterations in the species *Colossoma macropomum*, after contact with cunaniol in the water, varied from immobility, extreme excitability, loss of posture reflex, and muscle

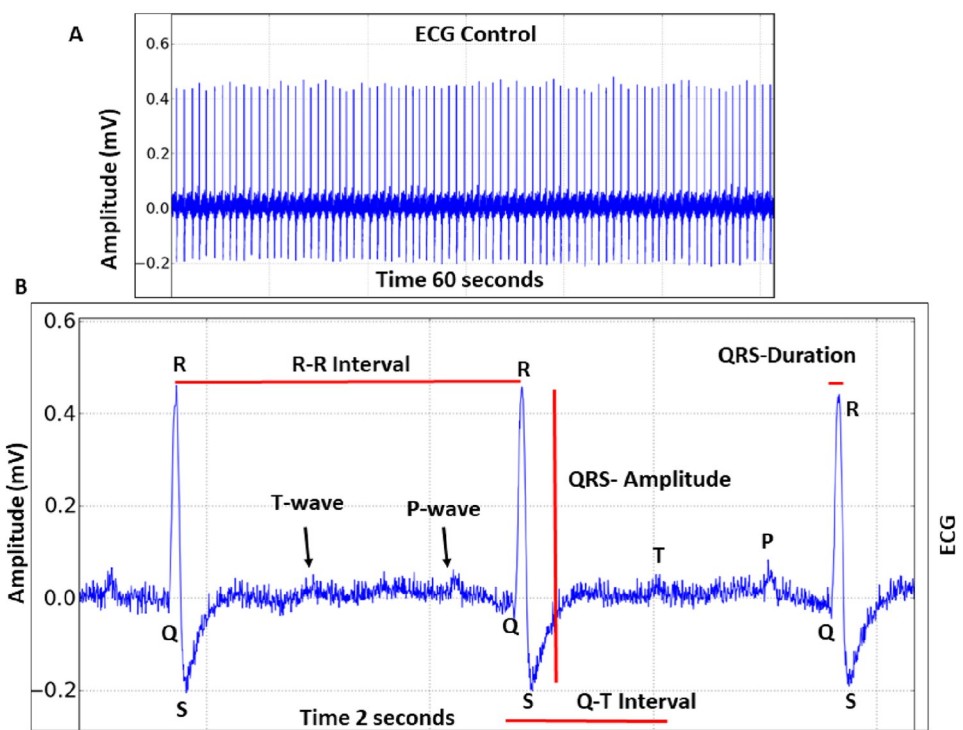

**Fig 5.** Electrocardiographic recording (ECG) (in the 60s) tambaqui control, *Colossoma macropomum* with cardiac deflagration demonstrating ECG amplitude (A) and enlarged tracing (in 2s) with the demonstration of P, T waves, QRS complex duration, and amplitude, Q-T and R-R intervals, (B).

spasms. Similarly, Quilliam and Stables (1969) [8] obtained the same result in goldfish and guppies, and Costa et al. (2006) [5] and Hamoy et al. (2018) [6] obtained central excitability behaviors in rats and mice. Other studies that analyzed the behavioral status of animals from experimental convulsive models achieved quite similar repercussions [9–12], showing that convulsive seizures caused by chemoconvulsants present similarities in behavior patterns.

Evidence of behavioral changes and the elucidation realized from electrophysiological analysis demonstrate the breakdown of homeostasis during contact with substances that alter organic functions [13]. The profile of the waves captured by the electroencephalographic record is subdivided into two main types, which are potential bursts, in which maximum amplitude and frequency are observed, and isolated spikes, characterized by a decrease in frequency and maintenance of the trigger amplitude, being these components found similarly in rats, although firing cycles were not observed [6]. The energy level during the potential bursts (Fig 1B) corresponds to the main element that causes the electrocorticogram to increase the linear scale amplitude (Fig 2).

The electromyographic investigation showed patterns that corroborate the immobility behavior, in which the amplitude of the register is low; the existence of an excitability phase in which amplitude and frequency are increased; and a phase of muscle spasms, which is characterized by rapid contraction, and at this time the animals show loss of posture reflex. Fujimoto et al. (2018) [14], when studying the anesthetic effect of clove oil in 3 Amazonian fish species, showed, in the electromyogram, the period of excitability with muscle hyperactivity and myorelaxation in the induction of anesthetic plan induced by the substance in fish. In induction, hyperactivity is observed followed by decreased swimming activity, loss of balance, and inability to respond to external stimuli [15–17]. Such similarity is so pronounced that there is

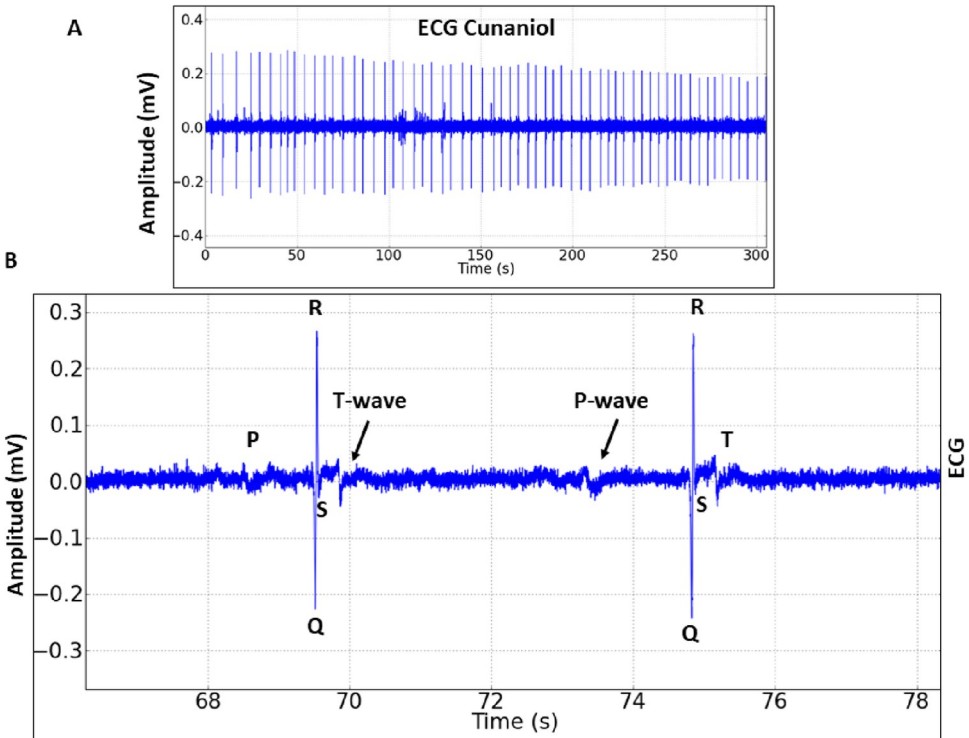

**Fig 6.** Electrocardiographic recording (ECG) of tambaqui, *Colossoma macropomum*, and its amplifications during bath with cunaniol at a concentration of 0.3 μg/L, corresponding to the final 300 s of contact with the cunaniol (A). Enlargement of ECG records in 12 seconds demonstrating the components of the record and heart in intense bradycardia (B).

another study that considers cunaniol as an anesthetic inducer [18], although the electrocorticographic recordings confirm the excitatory activity of the CNS. Due to the maintenance of EMG muscle spasms and isolated EEG shots, which are not compatible with anesthetic plans, the possibility of anesthesia is contradicted. Bradycardia could be identified by electrocardiogram, given the decrease in heart rates and amplitude. The increased time required to depolarize and repolarize reiterates this characteristic decrease in the contractile capacity of the heart in *Colossoma macropomum* (Table 1). In this respect, when related to the seizure picture, changes in cardiac activity in both animals [19] and humans can be evidenced as a result of the autonomic changes caused by seizures [20–24]. However, paradoxically, the occurrence of eventual tachycardias is similarly seen and the explanation for this dichotomy is still uncertain [25]. The effect on decreasing or increasing cardiac activity in fish may be related to an indirect action mediated by effects on the central nervous system, which during seizures can activate autonomous mechanisms that may explain these changes.

**Table 1. Comparison between heart rate means, in beats per minute (BPM), Amplitude (mV), RR interval (seconds), QT interval (seconds), and QRS duration (seconds) of tambaqui juveniles, *Colossoma macropomum*, between control and groups in contact with cunaniol 0.3 μg/L (*** indicates differences between control and treated groups indicate significant differences between groups [Test t and Mann-Whitney test (*p <0.05, **p<0.01, ***p<0.001. n = 9)].**

| Group / Cardiac Activity | Heart rate (bpm) | Amplitude (mV) | R-R interval (s) | Q-T interval (s) | QRS Duration (s) |
|---|---|---|---|---|---|
| Control | 96.22 ± 5.608 | 0.3909 ± 0.07593 | 0.6084 ± 0.02037 | 0.2675 ± 0.04457 | 0.03012 ± 0.001767 |
| Cunaniol | 23.11 ± 10.02*** | 0.3169 ± 0.03283* | 5.487 ± 1.456*** | 0.4271 ± 0.06760*** | 0.05858 ± 0.008562*** |

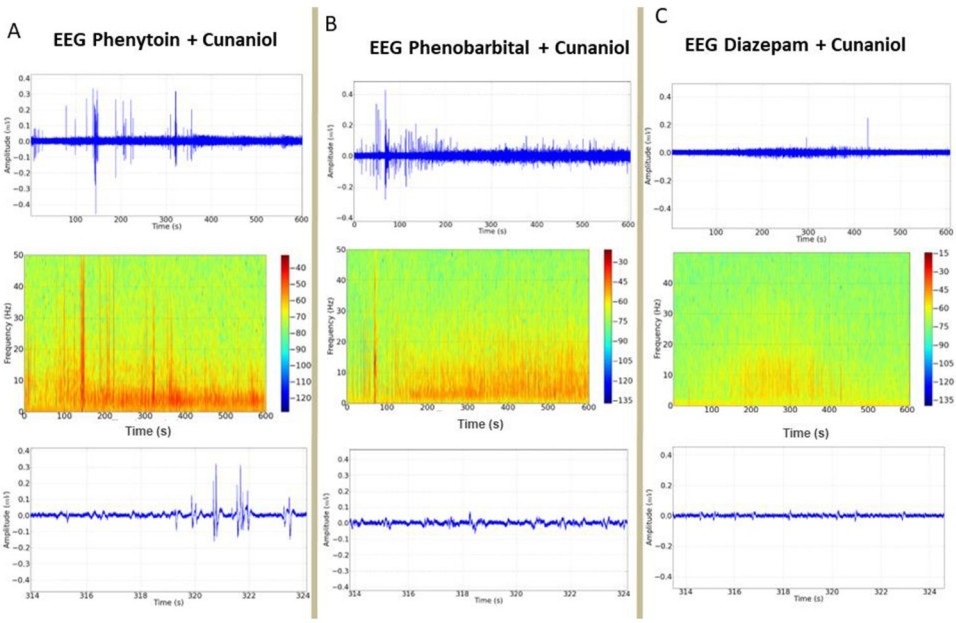

**Fig 7. Demonstration of the electroencephalographic (EEG) tracing of tambaqui (*Colossoma macropomum*), its spectrograms and recording amplifications, after application of phenytoin 10 mg/kg i.p.** (A); phenobarbital 10mg/kg i.p. (B) and Diazepam 10mg/kg i.p. (C) followed by an immersion bath with cunaniol to control induced seizures. Records lasting 600 seconds.

To evaluate the action of antiepileptics in the control of cunaniol-induced seizures, doses of diazepam, phenytoin, and phenobarbital were administered before contact with cunaniol. Observation of the electrocorticographic findings pointed out that diazepam was the drug that established the most effective in reducing the seizure activity of the chemical agent. Costa et al. (2006) [5] and Hamoy et al. (2018) [6] demonstrated similar results when studying these drugs' action in inhibiting the CNS's excitatory characteristics caused by the extract of leaves of the genus *Clibadium* and cunaniol isolated in mammals.

However, the administration of phenytoin was not effective in controlling seizures caused by cunaniol. Diazepam and phenobarbital have GABAergic agonist activity, which may support the hypothesis of cunaniol acting on GABA-A receptors [5]. Some studies have shown that cunaniol has non-competitive GABA-A receptor antagonist activity [4, 26]. Nevertheless, the full knowledge of its mechanism of action has not yet been fully achieved, which extends, therefore, the potential of pharmacological discoveries of this compound, that may broaden the spectrum of what is known today about seizures, epileptogenic processes, and epilepsy.

The method using electrophysiology to evaluate the toxicity of substances that act on the central nervous system has been increasingly used in fish, this tool helps to partially elucidate the mechanisms of action of certain drugs. Cunaniol triggers excitability in the central nervous system, which is reflected in high-potential muscle contractions, followed by loss of posture reflex. The heart showed a decrease in several parameters, causing cardiac depression in juveniles of *Colossoma macropomum*. Opening up opportunities for more targeted research in the future to elucidate microscopic and molecular actions

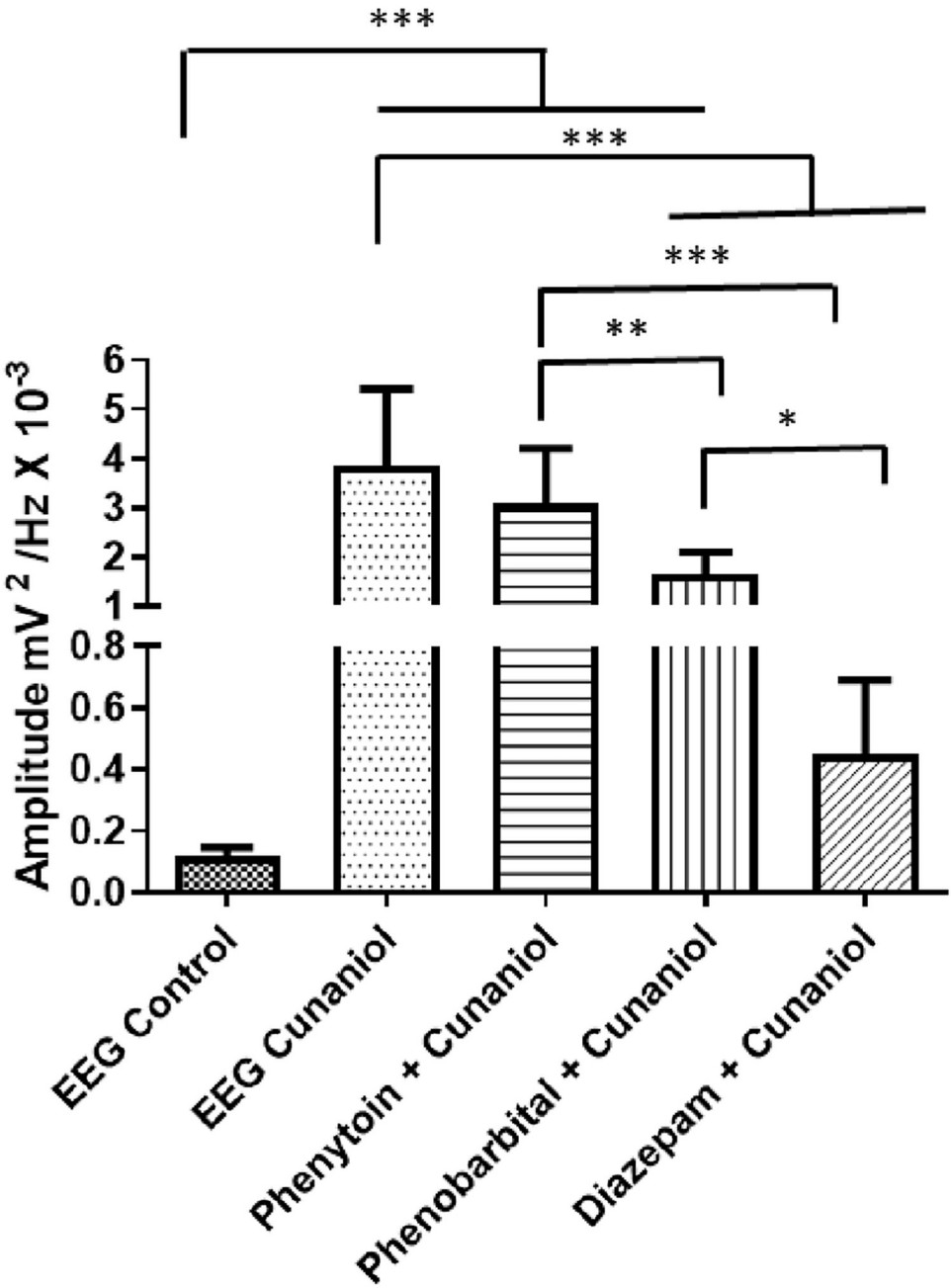

**Fig 8. Mean amplitude recorded during EEG of *Colossoma macropomum* juveniles submitted to doses of 10 mg/kg i.p.** of anticonvulsants Diazepam, phenytoin, and phenobarbital 10 minutes before contact with 3 μg/L cunaniol. Recordings were made at 600s duration with frequencies up to 50 Hz. [ANOVA and Tukey test (n = 12)].

## Supporting information

**S1 Dataset.**
(XLSX)

## Author Contributions

**Conceptualization:** Akira Otake Hamoy, Moisés Hamoy.

**Data curation:** Akira Otake Hamoy, Moisés Hamoy.

**Formal analysis:** Giovanna Lourenço Cei.

**Investigation:** Giovanna Lourenço Cei, Fábio Leite do Amaral Júnior.

**Methodology:** Fábio Leite do Amaral Júnior, Rafaela Marques Ribeiro.

**Project administration:** Suzane Maia da Fonseca, Moisés Hamoy.

**Software:** Maria Klara Otake Hamoy.

**Validation:** Rafaela Marques Ribeiro, Luis Andre Luz Barbas, Nilton Muto.

**Visualization:** Maria Klara Otake Hamoy.

**Writing – original draft:** Suzane Maia da Fonseca, Nilton Muto, Moisés Hamoy.

**Writing – review & editing:** Luis Andre Luz Barbas, Nilton Muto.

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
