## [Decision Letter · Decision Letter 0]

17 Feb 2023

PONE-D-22-30736Cunaniol: Behavioral, electrophysiological changes in Colossoma macropomum (Tambaqui)PLOS ONE

Dear Dr. Muto,

Thank you for submitting your manuscript to PLOS ONE. After careful consideration, we feel that it has merit but does not fully meet PLOS ONE’s publication criteria as it currently stands. Therefore, we invite you to submit a revised version of the manuscript that addresses the points raised during the review process.

We look forward to receiving your revised manuscript.

Kind regards,

Pan Li, PhD

Academic Editor

PLOS ONE

Journal Requirements:

Reviewers' comments:

Reviewer's Responses to Questions

**Comments to the Author**

1. Is the manuscript technically sound, and do the data support the conclusions?

Reviewer #1: Partly

Reviewer #2: Yes

2. Has the statistical analysis been performed appropriately and rigorously? 

Reviewer #1: N/A

Reviewer #2: Yes

3. Have the authors made all data underlying the findings in their manuscript fully available?

Reviewer #1: No

Reviewer #2: Yes

4. Is the manuscript presented in an intelligible fashion and written in standard English?

Reviewer #1: No

Reviewer #2: Yes

5. Review Comments to the Author

Reviewer #1: Dear authors,

The study of Cunaniol-induced seizures in fish is very interesting and holds great potential. I hope that the feedback provided below will enhance the quality of the manuscript and increase its chances of acceptance and publication.

Kind regards.

Comments:

Page 1, lines 1-2: “Cunaniol: Behavioral, electrophysiological changes in Colossoma macropomum (Tambaqui)”. I recommend modifying the title of the manuscript to make it clearer.

Page 4, lines 93-94: “In relation to its chemical composition, cunaniol C14H14O2 has as nomenclature: (2R, 3S) -2 - [(Z) -non-1-en-3,5,7-triinyl] oxan-3-ol] following IUPAC rules”. Please enclose the chemical formula of cunaniol in parentheses, and also provide the meaning of the IUPAC acronym (International Union of Pure and Applied Chemistry).

Page 5, lines 103-104: “… potent action of this extract as a chemoconvulsant agent with changes in brain oscillations in Beta (12 to 28 Hz)”. To make the text clearer, I suggest using ‘changes in Beta brain oscillations (12 to 28 Hz)’.

Please correct and standardize the bibliographic references in the text to conform to the journal guidelines for citations (https://journals.plos.org/plosone/s/submission-guidelines#loc-references). For example, in the body of the text, you can cite ‘Costa and colleagues (2006) [corresponding reference number here]…’ (page 5, line 105).

Page 5, line 116: “… to analyze behavioral, and electrophysiological responses in…”. The comma is not necessary.

Page 7, line 166-169: “… and the same tissues were collected for oxidative stress evaluation; (…) at the end of this experiment, muscle, brain and heart tissues were collected to evaluate oxidative stress after contact…”. The results of the experiments measuring oxidative stress are not included in the manuscript. Therefore, I recommend either removing this information from the text or providing an explanation for its intended publication in a future work.

Pages 7 and 8, lines 161-180: Please improve the presentation of the groups treated in each experiment.

Page 11, lines 255-261: Please clearly indicate the statistical analysis used in each performed experiment.

Page 12, line 275: Table 1 is redundant as its information is already detailed in the text.

Page 12, line 277: Replace “electroencephalographic record amplitude” with ‘EEG amplitude’.

Figure 1: In Figures 1A and 1B, there is a correlation between the left and right images (1A = 300 s; 1B = 600 s), hence they should be aligned closely. The images in the center are cutouts of the original tracing of the electrophysiological record. Therefore, it is recommended to place them to the right of the original image, which should have the selected sections clearly marked. In my opinion, Figures 1A and 1B should be arranged in a vertical orientation, with each figure having two columns (similar to Figure 3B), to enhance their visual appeal and make interpretation of the images easier.

Page 14, line 317: “The normal ECG of C. macropomum is shown in Figures 317 6 A and B, ...”. I believe that the normal ECG is displayed in Figure 5.

Page 15, line 355: “… induced seizure outbreaks, changes in the EEG recordings observed in Figure 8 C were…”. The correct figure is 7C.

Figures 3 and 7: I suggest redoing the figures based on the comment about figure 1.

Page 18, line 425: I suggest concluding the discussion with questions and perspectives on the use of this substance in future works.

Finally, I suggest a thorough review of the text to correct errors and improve coherence.

Reviewer #2: The authors should better describe the toxic substance Cunaniol in the introduction.

The authors do not make it clear, whether the effect of cunaniol on bradycardia is direct or indirect?

Cunaniol is an alcohol and could be blocking ionotropic glutamate receptors , why the authors did not test this possibility?

6. PLOS authors have the option to publish the peer review history of their article (what does this mean?). If published, this will include your full peer review and any attached files.

Reviewer #1: No

Reviewer #2: **Yes: **José Luiz Martins do nascimento

---

## [Author Response · Author response to Decision Letter 0]

2 May 2023

Response to Reviewer’s Comments:

Reviewer #1: 

The study of Cunaniol-induced seizures in fish is very interesting and holds great potential. I hope that the feedback provided below will enhance the quality of the manuscript and increase its chances of acceptance and publication.

Kind regards.

Comments:

Page 1, lines 1-2: “Cunaniol: Behavioral, electrophysiological changes in Colossoma macropomum (Tambaqui)”. I recommend modifying the title of the manuscript to make it clearer.

As the reviewer suggested, we modified the title of the manuscript. Line 1.

Page 4, lines 93-94: “In relation to its chemical composition, cunaniol C14H14O2 has as nomenclature: (2R, 3S) -2 - [(Z) -non-1-en-3,5,7-triinyl] oxan-3-ol] following IUPAC rules”. Please enclose the chemical formula of cunaniol in parentheses, and also provide the meaning of the IUPAC acronym (International Union of Pure and Applied Chemistry).

As the reviewer pointed out, we provided the acronym of IUPAC and the chemical composition of cunaniol. Line 94-95.

Page 5, lines 103-104: “… potent action of this extract as a chemoconvulsant agent with changes in brain oscillations in Beta (12 to 28 Hz)”. To make the text clearer, I suggest using ‘changes in Beta brain oscillations (12 to 28 Hz)’.

According to reviewer comments, we corrected “changes in Beta brain oscillations..”

Please correct and standardize the bibliographic references in the text to conform to the journal guidelines for citations (https://journals.plos.org/plosone/s/submission-guidelines#loc-references). For example, in the body of the text, you can cite ‘Costa and colleagues (2006) [corresponding reference number here]…’ (page 5, line 105).

Done.

Page 5, line 116: “… to analyze behavioral, and electrophysiological responses in…”. The comma is not necessary.

Done

Page 7, line 166-169: “… and the same tissues were collected for oxidative stress evaluation; (…) at the end of this experiment, muscle, brain and heart tissues were collected to evaluate oxidative stress after contact…”. The results of the experiments measuring oxidative stress are not included in the manuscript. Therefore, I recommend either removing this information from the text or providing an explanation for its intended publication in a future work.

As the reviewer suggested, we removed the information for oxidative stress evaluation

Pages 7 and 8, lines 161-180: Please improve the presentation of the groups treated in each experiment.

Done.

Page 11, lines 255-261: Please clearly indicate the statistical analysis used in each performed experiment.

Done. Line 258-260

Page 12, line 275: Table 1 is redundant as its information is already detailed in the text.

Page 12, line 277: Replace “electroencephalographic record amplitude” with ‘EEG amplitude’.

Done. Line 277.

Figure 1: In Figures 1A and 1B, there is a correlation between the left and right images (1A = 300 s; 1B = 600 s), hence they should be aligned closely. The images in the center are cutouts of the original tracing of the electrophysiological record. Therefore, it is recommended to place them to the right of the original image, which should have the selected sections clearly marked. In my opinion, Figures 1A and 1B should be arranged in a vertical orientation, with each figure having two columns (similar to Figure 3B), to enhance their visual appeal and make interpretation of the images easier.

As the reviewer suggested, we placed the spectral frequency in the middle.

Page 14, line 317: “The normal ECG of C. macropomum is shown in Figures 317 6 A and B, ...”. I believe that the normal ECG is displayed in Figure 5.

According to reviewer’s comments, we replaced the sentences according to Fig 5 to Fig 6. Line”

315 - 320

Page 15, line 355: “… induced seizure outbreaks, changes in the EEG recordings observed in Figure 8 C were…”. The correct figure is 7C.

Done.

Figures 3 and 7: I suggest redoing the figures based on the comment about figure 1.

Done.

Page 18, line 425: I suggest concluding the discussion with questions and perspectives on the use of this substance in future works.

Done.

Finally, I suggest a thorough review of the text to correct errors and improve coherence.

Reviewer #2: 

Comments:

The authors should better describe the toxic substance Cunaniol in the introduction.

Unfortunately, there are still few studies describing Cunaniol and its toxicity. The present study is focusing in elucidate better properties of this Amazonian plant. 

The authors do not make it clear, whether the effect of cunaniol on bradycardia is direct or indirect?

As the reviewer suggested, we added the information: “The effect on decreasing or increasing cardiac activity in fish may be related to an indirect action mediated by effects on the central nervous system, which during seizures can activate autonomous mechanisms that may explain these changes.” Line 402 – 405.

Cunaniol is an alcohol and could be blocking ionotropic glutamate receptors , why the authors did not test this possibility?

Our objective was to evaluate the blocking of the anticonvulsant effect with classic drugs (phenobarbital, phenytoin and diazepam). To compare the effects of cunaniol with a glutamatergic agonist. 

Previously, we published an in vitro study: “Evaluation of uptake glutamate in cultures of retina cells in the presence of cunaniol”. In that study, the cunaniol was able to block the uptake of glutamate in neuronal cell cultures, confirming that this effect was concentration dependent.

---

## [Decision Letter · Decision Letter 1]

13 Jun 2023

Behavioral, electrocorticographic and electrocardiologic changes in Colossoma macropomum (Tambaqui) in the effect of cunaniol

PONE-D-22-30736R1

Dear Dr. Muto,

We’re pleased to inform you that your manuscript has been judged scientifically suitable for publication and will be formally accepted for publication once it meets all outstanding technical requirements.

Kind regards,

Pan Li, PhD

Academic Editor

PLOS ONE

Additional Editor Comments (optional):

Reviewers' comments:

Reviewer's Responses to Questions

**Comments to the Author**

1. If the authors have adequately addressed your comments raised in a previous round of review and you feel that this manuscript is now acceptable for publication, you may indicate that here to bypass the “Comments to the Author” section, enter your conflict of interest statement in the “Confidential to Editor” section, and submit your "Accept" recommendation.

Reviewer #2: All comments have been addressed

2. Is the manuscript technically sound, and do the data support the conclusions?

Reviewer #2: Yes

3. Has the statistical analysis been performed appropriately and rigorously? 

Reviewer #2: Yes

4. Have the authors made all data underlying the findings in their manuscript fully available?

Reviewer #2: (No Response)

5. Is the manuscript presented in an intelligible fashion and written in standard English?

Reviewer #2: Yes

6. Review Comments to the Author

Reviewer #2: When resubmitting the manuscript, the authors

included a listing the points

raised by the reviewe, and including both a final

version of the paper and a highlighted version showing

where the changes have been made.

The authors have reviewed the manuscript extensively and is now appropriate to be published.

7. PLOS authors have the option to publish the peer review history of their article (what does this mean?). If published, this will include your full peer review and any attached files.

Reviewer #2: **Yes: **José Luiz Martins do Nascimento

---

## [Editor Report · Acceptance letter]

21 Jun 2023

PONE-D-22-30736R1 

Behavioral, electrocorticographic and electrocardiologic changes in *Colossoma macropomum* (Tambaqui) in the effect of cunaniol 

Dear Dr. Muto:

I'm pleased to inform you that your manuscript has been deemed suitable for publication in PLOS ONE. Congratulations! Your manuscript is now with our production department. 

Kind regards, 

on behalf of

Dr. Pan Li 

Academic Editor

PLOS ONE